# Genomic-Based Newborn Screening for Inborn Errors of Immunity: Practical and Ethical Considerations

**DOI:** 10.3390/ijns9020022

**Published:** 2023-04-11

**Authors:** Jovanka R. King, Kalle Grill, Lennart Hammarström

**Affiliations:** 1Department of Allergy & Clinical Immunology, Women’s and Children’s Hospital Network, North Adelaide, SA 5006, Australia; jovanka.king@adelaide.edu.au; 2Immunology Directorate, SA Pathology, Adelaide, SA 5000, Australia; 3Robinson Research Institute and Discipline of Paediatrics, School of Medicine, University of Adelaide, North Adelaide, SA 5006, Australia; 4Department of Historical, Philosophical and Religious Studies, Umeå University, SE-90187 Umeå, Sweden; kalle.grill@umu.se; 5Department of Biosciences and Nutrition, Neo, Karolinska Institutet, SE-14183 Huddinge, Sweden

**Keywords:** inborn errors of immunity (IEI), newborn screening, next-generation sequencing (NGS), whole-exome sequencing (WES), whole-genome sequencing (WGS), ethical, legal and social considerations (ELSI)

## Abstract

Inborn errors of immunity (IEI) are a group of over 450 genetically distinct conditions associated with significant morbidity and mortality, for which early diagnosis and treatment improve outcomes. Newborn screening for severe combined immunodeficiency (SCID) is currently underway in several countries, utilising a DNA-based technique to quantify T cell receptor excision circles (TREC) and kappa-deleting recombination excision circles (KREC). This strategy will only identify those infants with an IEI associated with T and/or B cell lymphopenia. Other severe forms of IEI will not be detected. Up-front, first-tier genomic-based newborn screening has been proposed as a potential approach by which to concurrently screen infants for hundreds of monogenic diseases at birth. Given the clinical, phenotypic and genetic heterogeneity of IEI, a next-generation sequencing-based newborn screening approach would be suitable. There are, however, several ethical, legal and social issues which must be evaluated in detail prior to adopting a genomic-based newborn screening approach, and these are discussed herein in the context of IEI.

## 1. Introduction

The aim of newborn screening programs is to identify infants with a range of significant conditions for which there is a pre-symptomatic phase and effective treatment is available. Since the description of phenylketonuria (PKU) in the 1930s and subsequent establishment of a laboratory assay to identify asymptomatic infants with this condition in the 1950s [1], there have been significant technological advances that have enabled an expansion of the number and range of treatable conditions identified through newborn screening (NBS) programs. Traditionally, newborn screening tests have centred around a tandem mass-spectrometric (MS/MS) approach, which is effective in identifying a range of conditions including inborn errors of metabolism, congenital hypothyroidism and congenital adrenal hyperplasia. Technological advances have driven the expansion of NBS programs to include a wider range of diseases [2], for example, severe inborn errors of immunity (IEI) such as severe combined immunodeficiency (SCID), using DNA-based technologies (measurement of T cell receptor excision circles (TREC) and kappa-deleting recombination excision circles (KREC) by quantitative PCR).

Inborn errors of immunity are a heterogenous group of disorders which manifest as severe, unusual or recurrent infections; immune dysregulation (including autoimmunity); and other clinical features. There are now over 485 different monogenic IEI which have been described, and this number continues to increase at a rapid rate [3]. Marked diagnostic delay (up to several years) and a long ‘diagnostic odyssey’ are experienced by many affected individuals. This results in delayed treatment and subsequent increased morbidity, mortality and poorer outcomes. SCID is a particularly severe IEI which presents in infancy and is fatal without definitive treatment with either allogeneic hematopoietic stem cell transplantation (HSCT) or gene therapy (GT). Outcomes are significantly improved if this is undertaken at an early age (preferably prior to 3.5 months of age) to avoid infections and other complications at the time of transplant [4]. This requires a diagnosis to be made in the first few weeks of life. In the absence of a known family history or prenatal diagnosis, this can only be achieved by screening infants in the neonatal period; hence, newborn screening for SCID has commenced in many countries throughout the world [5].

Current screening methodologies for SCID involve assays measuring TREC and/or KREC levels, which are surrogate markers for naïve T and B cell production and enable the identification of a range of IEI where T and/or B cell lymphopenia are a feature [5]. However, IEI are clinically, phenotypically and immunologically heterogeneous; thus, this screening approach will not capture all clinically relevant forms of IEI, including conditions such as neutrophil disorders, complement deficiencies and familial hemophagocytic lymphohistiocytosis (HLH). A range of alternative methodologies, including protein-based assays and copy number variant analyses, have demonstrated a proof of concept that screening for these conditions is possible [6,7,8,9]. However, employing a multitude of different methodological strategies for a range of conditions is not practically nor economically feasible. Given that IEI are genetically determined, an alternative approach would be concurrent, parallel screening of hundreds of disease-causing genes using next generation sequencing (NGS), employing either whole-exome sequencing (WES) or whole-genome sequencing (WGS) as an up-front, first-tier testing strategy. This challenges the current paradigm, whereby genetic sequencing is employed as a second- or even third-tier test in newborn screening algorithms. Despite disease heterogeneity, many forms of IEI have one commonality: an identifiable genetic target. This suggests that an NGS-based screening strategy is a rational approach, providing a single platform to screen for hundreds of diseases simultaneously.

An NGS-based screening approach is particularly suitable for the identification of infants with IEI based on disease heterogeneity, lack of a suitable biochemical marker to screen for all conditions simultaneously, and the genetic basis of this group of disorders. At the same time, we believe that there is further scope for its application in newborn screening for other conditions with a monogenic basis, including inborn errors of metabolism and a large range of other conditions. NGS provides a single modality which enables parallel screening for hundreds of different disorders which differ in terms of clinical phenotype and disease-specific biomarkers, making it an attractive option for newborn screening.

The feasibility of a first-tier, rapid WGS-based newborn screening approach has been demonstrated by Kingsmore et al., who identified 388 clinically actionable conditions in 2208 critically unwell neonates in Intensive Care Units with 99.7% specificity and 88.8% sensitivity [10]. Genetic screening of sick newborn infants rarely presents any ethical, legal or social concerns, and is currently regarded a routine medical service. Other studies have also demonstrated the utility of NGS in providing a definitive genetic diagnosis in patients with rare diseases, informing disease prognostication and enabling commencement of effective treatment [11]. This work has demonstrated proof of concept of NGS testing approaches to improve diagnostics and clinical care of acutely unwell infants. In particular, it has provided evidence for the application of this methodology and the provision of rapid results to facilitate time-critical clinical decision making. These findings may be extrapolated to NBS programs, which rely on rapidly available results for early intervention. This approach has been demonstrated to be an effective testing modality to screen for hundreds of different genes simultaneously. There are some differences, however, in that the majority of unselected newborns undergoing screening for disease will be ‘healthy’. Thus, first-tier screening of ALL newborns, as we are advocating, requires careful reflection. This screening approach has already been evaluated in a study of 321 unselected newborns in China, in whom first-tier WGS identified pathogenic or likely pathogenic variants associated with 59 Mendelian disorders in one-third of screened neonates [12]. There is a view to further evaluation in other, larger, prospective newborn screening programs, as is the case with the Genomics England Newborn screening project [13]. Table 1 highlights the findings of NGS-based screening studies to date, including both unwell infants and unselected newborns [10,12,14,15,16,17,18,19].

There are several considerations which must be made prior to adopting a first-tier NGS-based screening approach for IEI, spanning those of a practical nature (turn-around time, costs, clinical follow-up protocols, etc.) and methodological and technical factors (test characteristics and acceptability, quality assurance, bioinformatic pipelines and analysis, variant calling, etc.) [20]. NGS-based techniques include both WES and WGS, and the choice of methodology is based on a number of factors. The current standard clinical approach to genetic investigation of patients with a suspected IEI involves the evaluation of a ‘panel’ of IEI-associated genes by WES. This panel-based WES approach has also been used in the majority of NGS-based NBS studies to date. It is anticipated that over time, these panels will expand to include a broader number of genes in alignment with new gene discovery. WGS will further increase the diagnostic yield. Methodological options will, thus, likely evolve and change over time. Importantly, ethical, legal and social issues (ELSI) must also be considered and rigorously evaluated, as this forms an important part of the dialogue around genomic-based screening [20]. These factors will now be discussed in the context of newborn screening for IEI.

### 1.1. Selection of Disease Candidates for Newborn Screening Programs

Wilson and Jungner published their recommendations for population screening in 1968 [21], outlining criteria to guide disease inclusion in screening programs. There have been recent calls to update these criteria in the context of technological advances and new therapies [22,23,24,25]. These updated criteria were recently reviewed in the context of newborn screening for IEI, highlighting the need to both consider alternative approaches and increase the spectrum of screened diseases, particularly in the context of developing new and improved therapies [20]. A more recent set of criteria was subsequently published by the US Advisory Committee on Heritable Diseases in Newborns and Children [26].

A wealth of information is generated from genetic sequencing; thus, careful consideration must be given to the specific disease candidates and specific disease-associated genes which should be included in NBS programs. In the case of IEI, along with the identification of clear pathogenic mutations in disease-associated genes, there is potential for the discovery of variants of unknown significance (VUS), carrier statuses for diseases which may have relevance for individuals in their later reproductive years and adult-onset conditions. It is anticipated that each individual will be a carrier for one or more conditions. Some clinically heterogenous conditions may be associated with VUS or variants giving rise to sub-clinical disease. In addition, advances in knowledge may lead to the reclassification of variants previously assigned as VUS to pathogenic, with potential implications for the clinical management of individuals. All of these situations, as well as how they will be managed, need careful consideration, including the risk–benefit profile of disclosure (or non-disclosure) of various findings, which poses a challenge both ethically and clinically. It is likely that our approach will change over time as technological and methodological changes and increased knowledge is gained. At the current time, we would advocate that only those genes in which mutations are associated with clinically actionable IEI be included in newborn screening panels [20]. This is also supported by Johnston et al., who recommend screening only for select diseases in NBS programs for which treatment is available, and focusing only on pathogenic mutations [27]. With increased research and experience regarding this approach, and acquisition of knowledge over time, this will be refined. This will include modified disease and gene lists for interrogation, which will be regularly reviewed and updated in accordance with new findings, given that novel monogenic diseases will continue to be described and new therapies will become available. In addition, our ability to better classify and resolve VUS will also improve in the future.

There are now several hundred monogenic IEI which are broadly categorised into ten groups [28], but these are highly variable in terms of disease presentation, severity and available treatment options. These conditions, in the context of future NGS-based NBS programs, were reviewed in detail by King, Notarangelo and Hammarström in 2021 [20]. Although it was highlighted that all forms of IEI have potential interventions which will reduce morbidity and mortality and improve the quality of life of affected individuals, it was advocated that early genomic-based NBS programs should aim to identify infants with well-defined, significant and clinically actionable conditions which have effective treatments [20]. In the first instance, this would include conditions for which potentially curative therapies are available, including SCID, other combined immunodeficiencies and chronic granulomatous disease, which can be cured with successful HSCT or gene therapy. In addition, it was advocated that in the first instance, carefully selected and disease-associated genes be interrogated for known pathogenic mutations giving rise to clinical disease, with a view to expand both gene and variant lists over time in alignment with technological advancements, improved knowledge and increased experience with newly described IEI and IEI-associated variants [20]. There is a significant international effort underway to define, construct, curate and harmonise a comprehensive database of conditions and genes for IEI and a range of other conditions for diagnostic testing [29]. Applications such as PanelApp (https://panelapp.agha.umccr.org/) (accessed on 15 December 2022) are important for facilitating this collaborative process. Such efforts are especially important in the lead-up to NGS-based NBS pilot studies. These lists will no doubt continue to be critically analysed and further developed prior to implementation, and then regularly reviewed and updated once these programs are established.

### 1.2. Genetic Screening Will Enable Us to Identify a Wider Range of Clinically Actionable Conditions

The ethical, social and legal considerations of first-tier NGS-based screening are closely intertwined with the practical, methodological and technical factors of both the testing itself and the handling of genetic information during and after testing. In addition, our current (and future) medical knowledge determines which conditions can be identified, and with what level of certainty diagnoses can be made relative to the risk of false positives and false negatives. One particularly salient factor is whether NGS-based screening would be used as an exclusive first-tier approach or used in combination with current testing methodologies, based on biochemical and other markers. In the case of IEI, TREC/KREC enumeration is the only current, routinely available screening assay, and would only identify a small proportion of the many forms of IEI. Conversely, some infants returning an abnormal TREC and/or KREC result would not have an identifiable monogenic disease. As such, if used in combination with current techniques, genomic testing will both broaden the range of identified diseases and improve precision. Similarly, there are other conditions in addition to IEI that would not be identified by genetic screening, but would be found by screening based on biochemical or other markers. Given this fact, it remains to be seen whether first-tier NGS-based screening will replace current methodologies, be used where current methodologies do not exist (e.g., for diseases where there is no current MS/MS platform) or be used in tandem, and whether these factors may need to be different for specific diseases in order to increase diagnostic yield. A recent evaluation of MS/MS versus NGS screening approaches for inborn errors of metabolism suggested that the former is superior at the current time in terms of sensitivity and specificity [30], although this suggestion has been challenged [10,12]. This issue requires further evaluation in the future once results of larger, prospective studies evaluating first-tier genomic-based NBS are available. From the perspective of IEI, aside from those conditions identifiable by TREC/KREC screening, as aforementioned, there is no alternative testing strategy currently available. Thus, NGS-based testing would hence be the only option, aside from the few diseases where MS/MS testing may be a possibility (ADA deficiency and purine nucleoside phosphorylase (PNP) deficiency) [31,32].

## 2. Ethical, Legal and Social Implications (ELSIs) in Screening for IEI

We can distinguish a number of ELSIs that will be relevant to different extents for different screening approaches, including NGS-based NBS for IEI. Many of these are further discussed in detail by Johnston et al. [27]. Herein, we have divided the key ELSIs into two main categories: the relatively direct effects of testing on the individual level, and the more indirect or conditional effects that depend on what solutions are found at the societal level. We will now address these two categories in turn in the context of IEI, while distinguishing within those categories the major, more specific issues, as we see them.

### 2.1. Individual: Physical and Psychological Well-Being

Genomic screening will enable the diagnosis of conditions that would be missed using current testing modalities. As such, this approach will result in both an increase in the breadth of different IEI identified at birth and an increase in the overall number of cases identified. The resulting improvements in health and physical well-being represent the obvious rationale for introducing first-tier genetic screening. More children will be diagnosed with IEI and receive effective treatments earlier. In addition, in the context of pharmacogenomics, the potential for adverse reactions to medications may be identified prior to drug administration, which will enable the prevention of predictable and avoidable adverse outcomes. If the testing method is such that it avoids false negatives, there is also an informational benefit for those who are found not to have any actionable conditions. On the other hand, with a risk of false negatives comes not only the risk of being misinformed about disease susceptibility, but also the risk that a particular condition, such as SCID or XLA, will be under-diagnosed because it is known to be included in a screening program and, thus, physicians may be less aware of it as a differential diagnosis.

Unlike some other screening programs, NBS testing itself involves little medical risk, given that it is currently based on a minimally invasive heel prick. However, health and well-being may be negatively affected by the receipt of an abnormal screening result which may be suggestive of an IEI. Firstly, the result can cause anxiety, particularly for the parents at the time of testing. This risk is greater if parents receive poor pre-test information and post-test counselling, potentially causing misunderstanding or confusion regarding the test results. For example, although the target condition for TREC screening is SCID, an abnormal initial screening result may, in fact, normalise upon re-testing or be of unclear clinical significance for that particular infant. Hence, the weight of this ethical consideration depends, to a very large extent, on the quality and accessibility of both pre-test information and genetic counselling. Secondly, the result can lead parents to request treatment and for physicians to provide it even though it is not warranted. In the words of Lund et al., there is a risk of ‘reporting of benign variants or variants for late-onset diseases, leading to unnecessary medicalising of the child, giving unnecessary treatment and creating patients-in-waiting’ [33]. Overtreatment can be harmful to the individual and is costly to society.

The balance of both the health benefits and the psychological and social risks will depend on what disease candidates are selected for screening. This selection, therefore, must be a considered process [34]. It is almost universally accepted that target conditions should have an effective treatment option available. It can be argued that all IEI have potential management options, spanning empirical and targeted therapies which may be preventive, supportive or curative [20]. For example, a patient identified to have an IEI affecting JAK-STAT signalling pathways such as *STAT1* or *STAT3* gain-of-function mutations can be offered targeted therapies abrogating abnormal signalling, and those with activated phosphoionositide 3-kinase delta syndrome may be offered specific therapies with mammalian target of rapamycin (mTOR) inhibitors or newly developed targeted small-molecule drugs [20]. The future will see the description of further new genetic mutations and the development of novel, improved targeted therapeutic options, further justifying a broader approach to NBS for IEI.

However, even with current screening programs, non-target conditions without treatment options are being identified, and this should also be considered in the pre-testing information offered to parents. An example of one important, non-target condition identified in SCID screening programs is ataxia–telangiectasia (AT), a condition presenting with progressive neurological decline in early childhood, variable immunodeficiency and reduced life expectancy. There is no curative treatment available at the current time. This has raised the question of whether it is more ethical to inform families of the diagnosis at the outset, as compared to allowing them to experience a typical infancy without disclosure of the diagnosis before the onset of symptoms as toddlers. When Blom et al. interviewed the families of healthy newborns enrolled in the Dutch SCID screening program, eighty-two percent of families indicated that early diagnosis was preferred, based on factors such as options for commencement of early supportive treatment and avoidance of diagnostic uncertainty [35]. These results highlight the importance of not making assumptions about families’ preferences, and reinforces the need to carry out further studies evaluating similar questions. Individuals are likely to hold different views, and perhaps information about non-actionable conditions could, to some extent, be individualized based on parents’ preferences, which could be registered during pre-delivery visits to the clinic.

Genotype–phenotype correlations in IEI are not uniform, and any selected genetic candidates for screening should be well-defined in terms of clinical features. Other factors such as penetrance must be well-understood [20]. Screening for less well-defined conditions should be avoided, so that only clearly pathogenic variants are reported. Concerns have been raised regarding the management of variants of unknown significance in terms of the risk of potentially causing undue anxiety in families, as well as the wider systemic implications (particularly in terms of workforce burden and cost) of second-tier testing and follow-up. It is anticipated that bioinformatic pipelines and variant-calling will be further refined over time, and reference can be made to population-specific databases to minimise over-calling of non-pathogenic variants. Provision of adequate pre-test information and information regarding the testing strategy may also help to minimise parental anxiety regarding follow-up clinical assessment and further testing.

It is imperative that newborn screening tests are acceptable to families in order to maintain high levels of uptake of these programs. All of the considerations discussed above can potentially impact the overall acceptability of an NGS-based NBS approach, and overall acceptability requires formal evaluation prior to implementation.

### 2.2. Individual: Autonomy and Informed Consent

The majority of newborn screening programs function on an opt-out basis, where testing will proceed routinely unless families decline it. For some ethicists, the fact that testing is proposed by society presents a threat to the autonomous choice of individuals, or, in the case of newborn screening, their parents [36]. Unlike testing that is actively sought out, screening imposes a pressure to conform because society has deemed testing not just worthwhile to provide, but important enough to initiate. From this perspective, expanding newborn screening for IEI increases the imposition on individual choice. The standard way of mitigating threats to autonomy in medical and many other contexts is, of course, informed consent. If people understand what they are being offered, and that they can opt out of testing without any further adverse consequences, then their voluntary acceptance of the offered testing will go a long way to mitigate any imposition. If one accepts this informed consent approach, then the radical expansion of newborn screening to cover many more conditions, including an increased number of IEI, as well as any increased probability of receiving information about carrier status, may seem problematic. This is because of the much more complex information that becomes relevant to making an informed choice either to accept the testing offered or to opt out [37,38].

However, this concern is arguably misdirected for several reasons. The first factor to note is that NBS is not a standard case of patient consent. The benefactor of newborn screening is primarily the newborn child, who has no capacity to give informed consent. Therefore, there is no basis for requiring outright consent. What might be possible is some sort of hypothetical or proxy consent (typically from parents). However, for individuals who are not and have never been competent to make decisions for themselves, society should arguably adopt a *best interest* approach and aim to further their interests rather than arrange for a substitute decision-making proxy to choose for them, as if they were merely temporarily incompetent [39]. An alternative would be to delay screening until adulthood, when individuals can provide meaningful consent, but at that point it would, of course, be too late to identify and treat many IEI, the majority of which have an early onset. This would be contrary to the overall aims and principles of NBS.

Parents are often considered the legitimate guardians of their children’s health in everyday contexts. However, this is arguably a pragmatic solution rather than a principled moral order. There is no general moral right for parents to make medical decisions for their children. This is reflected in the fact that in most jurisdictions, at least those that are somewhat liberal, the government will assume guardianship, to the exclusion of parents, if parents make important decisions that are contrary to the best medical interest of their children. It is, therefore, highly doubtful that any moral or legal right of parents to refuse newborn screening for IEI for their children should be based on respect for *patient* autonomy. Instead, there are arguably two main reasons for such a right: (1) to avoid alienating parents from the health care system, and (2) to respect parents’ general right to decide for their children, which prevails in more everyday matters [40,41]. In family ethics, it is a mainstream position that all rights of parents to decide for their children are ultimately based on the child’s interest in being parented, rather than in any interest or sui generis moral right of the parents (see Brighouse & Swift [40]). An argument for why parents make informed choices regarding NBS was presented by Nicholls [41]. If parents do not have the moral right to refuse testing of their children for serious immunological conditions, then they need not understand the implications of testing for that reason. However, this argument may not convince everyone. Furthermore, since we do allow parents to opt out, and since we want to preserve the often very high participation in newborn screening for IEI and other conditions, it is important to accommodate any expectations on the part of parents to be informed, or to be treated *as if* they had a moral right to make an informed decision. Doing so will increase acceptance. Hence, it is important to consider whether expanding screening to increase the breadth of detectable IEI will undermine the parental opportunity for informed choice.

In current screening programs, while information about the tests and the conditions tested for is typically available, there are generally modest (if any) attempts to ensure that parents have actively obtained and understood this information (there are variable approaches to pre-screening information provision in different settings [42]). This is not surprising, since the medical details of current screening programs are very difficult to understand and evaluate. In the case of IEI, there are a multitude of different conditions which are highly variable in terms of aetiology, clinical presentation and severity. Typically, in NBS programs, many diverse conditions are tested for, and each condition has a different testing method with varying probabilities for false positives and varying potential for second- and sometimes third-tier testing. In addition, given the complexity of the conditions themselves and the very small probabilities of being affected, to evaluate the overall risks and benefits of such tests is beyond the ability of most parents.

Against this background, it is not clear that an increase in complexity would change the ability of parents to make informed choices. Arguably, the type of information that parents might reasonably expect is not about the details of the testing methods (e.g., biochemical or genetic), nor about which IEI conditions might be identified and their pathogeneses. Rather, information that will be sought includes reassurance that the testing itself is harmless, details regarding the general purpose of screening (to identify very rare, but serious, IEI that can be treated) and perhaps the general possibility of false positive and false negative test results. The provision of this kind of information does not become much more difficult or expensive with expanded screening, nor with a change in methods. We advocate that pre-screening information should be provided to parents as part of an integrated, routine pre-natal care service. The level of information provided could potentially be tailored to the individual family based on their wishes, which could be recorded during their initial clinical encounters.

As highlighted, the approach to obtaining consent for newborn screening raises several ethical and practical issues, with many complexities that warrant consideration, particularly in the context of introducing genomic-based screening methodologies for IEI and other conditions. Importantly, trust in NBS programs is essential [34], and families can be reassured that the programs are well-thought out and established by governments after extensive planning, incorporating the involvement of many stakeholders and advice from experts. There are a multitude of safeguards in place to ensure that only relevant testing is performed.

### 2.3. Society: Privacy and Protection of Genetic Information

A broad variety of genetically determined IEI can be identified by genetic sequencing, and, as such, are potential candidates for genomic-based screening to identify these conditions in the newborn period. Genetic screening necessitates the processing and, at least for some time, storage of genetic information. This raises ethical, social and legal issues because genetic information is very rich and very personal. A person’s genome is a unique identifier, and extensive information can be extracted from DNA. In addition, the familial aspects of genetic data need to be considered, as the information may be relevant for other family members. This is the case for various IEI which may be caused by familial or de novo mutations, with variable modes of inheritance. This portends greater risks involved in the handling of genetic information, with a wider-reaching impact than may be initially anticipated. This also means that genetic information would be, to a large extent, collective, and so it makes sense to seek collective solutions for its handling.

It is crucial to develop testing procedures that protect genetic information. One safety measure would be to destroy all genetic information after analysis, keeping only the list of screened conditions given to the individual and their parents. However, there are potential benefits of long-term storage of the genetic information itself. In the case of IEI, there are ongoing, frequent new gene discoveries, and our understanding of the pathogenicity of genetic variants is always improving. As such, there may be benefits of having stored genomic data available for re-analysis at a later date in light of new information and knowledge. In addition, the evolution of new symptoms suggestive of an IEI in an individual could trigger rapid re-analysis of stored genomic data, reducing diagnostic and treatment delay. It does, however, remain uncertain whether retrospective access to stored genomic data will provide a more efficient, cost-effective process with a higher diagnostic yield compared to prospective indication-based re-sequencing based on clinical need. Firstly, data storage is expensive, and future re-sequencing may potentially be more cost-effective using newer techniques. In addition, technological advances may also result in improved diagnostic capabilities (for example, the ability to identify somatic variants, which underlie some forms of IEI and are not well-captured on current NGS platforms).

It is not medically justified to provide the parents of newborns with all information that may be relevant to the child during their lifetime. On the other hand, some information will be important later in life. As proposed by Biesecker et al., it would be ideal from a health perspective to disclose findings to individuals at a life stage at which interventions are beneficial, in relation to our best scientific knowledge at that time [43]. In the case of IEI, many of which present in infancy and early childhood, the provision of screening and diagnoses is most beneficial if performed early in life. As new conditions are identified and new treatments are developed, existing genetic information could be re-analysed and used to provide new and actionable information to individuals that was not accessible when they were infants, and that may potentially be very important at that later time (Chan and Petros 2019, reviewed by Biesecker et al.) [43]. This may also be applicable to pharmacogenomics, for which an individual’s genomic data could be re-visited just prior to the commencement of specific drugs to help predict the risk of adverse events and guide judicious prescription [43]. Although the focus of current genomic-based screening is the identification of monogenic diseases, future advances may also assist in identifying polygenic disorders and help to establish disease risk profiles [43].

On occasion, health systems have given private corporations access to genetic information [44]. On other occasions, law enforcement have been given access to blood samples obtained by population screening [45]. Even if such intentional breaches of confidentiality are avoided in the future, there are also risks of unintentional breaches. Any database that is connected to the internet can be hacked. One relevant example is the hacking of My Heritage, a company that sells genetic testing to individuals for genealogical purposes [46]. People working with genetic databases may make errors or be corrupt. Strict data management strategies are required to safeguard individuals from these risks, and must be maintained in order to minimise them. Proposed models are under evaluation using encryption approaches to safeguard genetic data in order to develop safe, secure data storage and sharing [47].

Another potential concern which has arisen is the impact of genetic screening results on interactions with agencies such as insurance providers, where a known genetic diagnosis of an IEI or other condition might have an impact on access to health insurance. While risks to individuals from insurance complications must be considered, we arguably cannot, as societies, allow the practices of insurance providers to determine how we deal with predictive and preventive medicine. Insurance practices and regulations should adjust to public health policy rather than the other way around. Different jurisdictions have, in practice, taken different approaches to this issue with various degrees of success [48]. Regulations should align with local needs and legal and social norms, as well as with general ethical considerations [49]. Although this may be of concern to some members of the public, in practical terms, this has not proven to be a significant issue, given that many countries have protective legislation in place (such as the Genetic Information Non-Disclosure Act in the US) which precludes the use of genetic information for insurance and other purposes.

When considering the privacy risks of genetic screening, we must also consider the alternatives. If a society does not implement population-wide genomic newborn screening for a range of conditions, including IEI, people may increasingly seek unregulated private or external screening where genetic information may be handled by disparate private actors to a larger extent, and the differences in what information is available for different people will vary depending not only on their own choices, but also on the choices of their relatives. Given the real risks associated with the dissemination of genetic information, it may be preferable that this information be handled by healthcare systems with clear recommendations which uphold regulatory requirements.

### 2.4. Society: Health and Economic Considerations

Rational and responsible expenditure of the healthcare budget is paramount. It is an ethical and social consideration from the perspective of ensuring that funds are used appropriately and will be of benefit to the community. Formal evaluation of health economic data and cost–benefit analyses for NGS-based NBS for IEI must be undertaken, and should take into account all aspects of NBS program-associated costs, including pre-test information provision, the testing itself, clinical follow-up, further testing and treatment [20]. We are seeing a steady decrease in the costs of genetic sequencing, owing to factors such as widespread uptake and technological improvements which have made NGS more readily available and competitively priced. Similar to the experience with screening for SCID using TREC/KREC analysis, where cost–benefit analyses have been in favour of screening [50,51,52], we anticipate that early diagnosis of patients with additional forms of IEI will ultimately decrease health-related expenditure. In the case of SCID, these studies have demonstrated that healthcare expenditure was reduced in infants diagnosed and managed early with HSCT, owing to a reduction in the utilisation of resources, including hospital and intensive care unit admissions and costs incurred in the management of serious disease complications. In the case of genomic-based screening for IEI, it is anticipated that the savings will likely outweigh any additional costs from overtreatment due to false positives and variants of unknown significance, especially over time, as healthcare professionals adjust to frequent use of genetic information. Rapid WGS for unwell infants in intensive care units has been shown not only to improve patient outcomes, but also to reduce healthcare expenditure [53]. In addition to cost-effectiveness studies, which directly assess healthcare expenditure at a systemic level, another important health economic consideration relates to Quality of Life Years (QALY) gained through early diagnosis and treatment. These health and economic considerations should be carefully evaluated in the context of NGS-based NBS.

Establishment of any new NBS program or a new approach to NBS, as discussed herein in the context of IEI, is a considerable task that relies not only on robust testing methodologies and pipelines, but personnel and resources to carry out the program effectively. This includes healthcare providers providing pre-test information and post-test counselling to parents, laboratory personnel processing specimens and performing sequencing analysis and clinical input from physicians for follow-up on infants returning abnormal screening results. There are many layers to a successful NBS program, as well as associated costs, which all need to be rigorously evaluated.

### 2.5. Society: Equity and Access to Newborn Screening

At present, there is wide variability in screening methodologies and screened conditions in different regions, even within the same country. Decisions regarding the addition of new candidate diseases, including IEI, to screening programs are multifactorial and may include considerations such as the population prevalence of specific diseases. In some settings, newborn screening is funded by public health systems and is free of charge to families, whereas in other areas, testing may incur a cost, which may impact upon uptake. These factors all impact the equity of testing and access to NBS, which is another important consideration when considering disease inclusion and the methodologies used in NBS programs.

An additional consideration in the implementation of genomic-based NBS for IEI are reference genome datasets. Current databases are predominantly based upon individuals from Northern European populations, and are, therefore, not representative of other ethnicities [20]. As such, it will be essential to build genomic population databases which are highly representative of different ethnic groups in order to enable robust variant calling and interpretation [20].

As noted above, healthcare screening should be compared to the alternative. If testing is a market good provided by commercial companies, access will depend on ability to pay as well as on education, access to information and other social factors. Commercial testing will also more likely lack robust clinical processes and genetic counselling capabilities, potentially resulting in less judicious testing with lower clinical utility [38]. This is particularly the case for IEI, as there is significant complexity owing to disease heterogeneity, requiring specialist clinical oversight of test interpretation and reporting. This, once again, reinforces the need for careful regulation of NBS and for NBS services to be linked with the appropriate clinical services to provide follow-up and treatment for any abnormal results received.

## 3. Conclusions

Up-front, first-tier genomic-based newborn screening shows promise for the identification of infants with IEI, a group of conditions for which there is significant clinical and phenotypic heterogeneity, through enabling the concurrent analysis of hundreds of genes. There are many factors which must be taken into account prior to adopting this approach to screening for IEI relating to practical, methodological and technical aspects, and, importantly, a range of ethical, legal and social considerations which must be fully evaluated. In particular, assessing the acceptability of the testing strategy is imperative to avoid undermining existing successful systems with very high uptakes, and we have discussed some of the issues in this context. Ethical, legal and social issues sometimes interact in non-obvious ways. While informed consent from parents may not be needed to safeguard autonomy, since newborns are not autonomous and so should rather have their best interest protected, it is important to provide parents with both information and the ability to opt-out so as to preserve the very high uptake. As technology advances, we are likely to see ongoing evolution in the approach to genomic newborn screening for IEI, starting with an initial, limited WES gene panel which will expand over time to include more genes. progresses to This is expected to then be replaced by WGS in order to improve diagnostic yield and expand screening capabilities for this broad group of disorders. Ongoing evaluation of all of these factors is essential in the planning for and implementation of genomic-based approaches to newborn screening for inborn errors of immunity and other clinically significant conditions.

## Figures and Tables

**Table 1 IJNS-09-00022-t001:** Outcomes of published studies evaluating rapid, first-tier NGS for both unselected (healthy) and unwell infants.

Author	Methodology	Cohort	Number of Screened Infants	Number of Genes Interrogated	Mendelian Diseases Covered	Key Findings	Reference
Jian et al., 2022	NBS WGS	Unselected neonates	321	251	59	136/321 (33.33%) pathogenic/likely pathogenic/copy number variants identified	[12]
Kingsmore et al., 2022	Simulated NBS rWGS	Critically unwell infants Biobank (healthy) subjects	2208 454,707	317	388 388	Negative predictive value 99.6%, sensitivity 88.8%, sensitivity 99.7%	[10]
Cho et al., 2017	Targeted NGS computational exome analysis, exome analysis	Unwell infants Controls (mutation carriers) (negative controls)	Total: 103 81 22 (10) (12)	307	159	5/25 (20%) known causal mutations in databases 20/25 (80%) rare variants (SNVs, nonsense mutations, short indels, gene duplication or deletion) 7/25 (30%) compound heterozygosity 93% sensitivity for core metabolic conditions	[14]
Bhattacharjee et al., 2015	WES ‘NBDx’ gene panel, in silico gene filter	Infants with known genetic disorders	36	126	27	75% sensitivity for core metabolic conditions	[15]
Bodian et al., 2016	Research-generated WGS data	Unselected infants	1696	163	28	88.6% true positive and 98.9% true negative rates for state-screened disorders Good genetic coverage of disorders using WGS	[16]
Willig et al., 2015	Rapid trio WGS	Unwell infants, <4 months of age, NICU/PICU	35	5430	20	20/35 (57%) genetic diagnoses achieved	[17]
Pavey et al., 2017	Trio WGS	Healthy cohort (genotype-first) Re-analysis of cohort with suspected IEI (phenotype-first)	1349 29	329 (IEI-associated genes)		396 (29%) pathogenic/likely pathogenic mutations identified 1/1349 (0.07%) clinically actionable IEI 3 (10%) non-IEI genetic diagnosis	[18]
Ceyhan-Birsoy et al., 2019	Singleton WES (trio re-analysis selected cases) BabySeq Project	Healthy newborns Unwell newborns in NICU	Total: 159 (127) (32)			15/59 (9.4%) risk of childhood-onset disease (10 from healthy cohort, 5 in NICU) 3/85 (3.5%) actionable adult-onset disease 140/159 (88%) carrier status for AR conditions 8/159 (5%) pharmacogenomic variants	[19]

WES: whole-exome sequencing, WGS: whole-genome sequencing, rWGS: rapid WGS, IEI: inborn errors of immunity, NICU/PICU: neonatal/paediatric intensive care unit.

## Data Availability

Not applicable.

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
