# Peer review of "Genomic-Based Newborn Screening for Inborn Errors of Immunity: Practical and Ethical Considerations"

_2409-515X, 2023, doi:10.3390/ijns9020022_

Round 1

Reviewer 1 Report

This manuscript addresses, as the title indicates, genomic-based newborn screening for inborn errors of immunity with an emphasis on the practical and ethical considerations.  Many of the arguments are familiar for NBS in general but the topic remains timely.  

The paper is focused on inborn errors of immunity, presumably because this is the expertise of the lead author.  There are no arguments made that this set of conditions is uniquely appropriate for genome-based screening.  The same article could be written for many other conditions or types of conditions.  The authors may want to add a few sentences regarding whether they think this class of conditions should be in the vanguard of such screening or whether it is one example among many.

I did not find it particularly helpful to see the literature review that includes genomic testing of symptomatic newborns.  This article is addressing population-based newborn screening so the testing of symptomatic newborns is not particularly relevant.

On page 6, the authors refer to the Wilson and Jungner criteria. This article is written in an international context but perhaps the most recent set of criteria for determining newborn screening conditions is those of the US Advisory Committee on Heritable Diseases in Newborns and Children.  (See: https://www.hrsa.gov/advisory-committees/heritable-disorders/decision-matrix)

A key concern is the inconsistent treatment of the issue of informed decision-making by parents.  On page 7, the authors state regarding the risk of anxiety and misinformation, "Hence, the weight of this ethical consideration depends to a very large extent on the quality and accessibility of genetic counseling." Yet on page 10, they acknowledge that programs make only "modest attempts" to ensure that parents have obtained and understand the information.  Even the latter is an overstatement.  Most parents are barely aware that newborn screening has been conducted and the brochures delivered about NBS fall very far short of genetic counseling. The authors need to argue that either the concerns over anxiety and misinformation are overstated or unimportant, or that the education and counseling approaches need to be drastically improved to accommodate genome sequencing.

The protection of genetic information is very important but the section on insurance discrimination could be significantly shortened.  The Genetic Information Non-disclosure Act (GINA) in the US precludes use of genetic information in most insurance and employment situations.  I suspect Australia and Sweden have similar protections, perhaps due to social medicine programs.  At any rate, this issue is a concern for patients and the public but it has not proven to be a real issue in practice and there is protective legislation in place in some countries.

I personally find the arguments about the reanalysis of stored genomic data to be unconvincing, although, as the authors note, the possibility is raised frequently in the literature.  I find it much more likely that a child or adult will obtain a new sequence using more accurate and comprehensive technologies of the future when and if there is a clinical indication.  The current section is largely fine but the authors might add the notion that it remains uncertain whether storage and later access of genomic data will prove to be more accurate and efficient than indication-based testing when a particular need arises.

The sentence on page 12, "We predict that QALY gained through more comprehensive, genomic NBS programs will be substantial, a benefit that should be included in the context of health economic analyses."  The point of these analyses is to determine the magnitude of the benefits and their cost so the sentence does not make sense and it may not be appropriate for non-health economists to make such predictions.  A call for careful economic analyses is certainly appropriate.

The last paragraph is inconsistent.  It is not appropriate to conclude in the first sentence that genomic based NBS is an ideal approach to these conditions but then acknowledge that there are a host of issues that need to be investigated and addressed. The authors might say that this approach is promising and the range of issues noted should be investigated.

An important set of issues that is not adequately addressed is the wealth of information that flows from sequencing.  How will information about carrier status and adult onset conditions be managed?  The ACMG list of actionable variants will be identified far more frequently than IEI conditions.  Each individual will be a carrier for one or more conditions.  The ethical challenge will be to address the management and disclosure, or non-disclosure, of this type of information.  Further, how should current variants of uncertain significance be addressed when future information determines that they are clinically relevant?  For clinically heterogeneous conditions, should programs disclose likely mild or subclinical variants?  I would like to see the authors look more closely at the spectrum of IEI conditions and grapple with the ethical issues that arise from what we know, and do not know, about the genetic underpinnings of those conditions.  Are there adult onset conditions and carrier states that would be detected and how should those results be addressed in the context of NBS.

The management of information relates to my last comment.  The authors are not, apparently, individuals working within public health departments that conduct NBS.  Major drivers of the complexity and cost of programs are not so much the tests per se, but the personnel necessary to generate, analyze, communicate, and follow-up the information generated.  Many programs (in the US) are overburdened and struggle to bring on any new test.  The authors need to acknowledge the monumental task entailed in their recommendations from the counseling/informing of parents through to the primary care physicians who will be receiving this complex information.

The authors may want to explicitly address some of the issues raised by this working group:

Johnston J, Lantos J, Goldenberg A, Chen F, Parens E, Koenig BA, NSIGHT Ethics and Policy Advisory Board. Sequencing Newborns: A call for Nuanced Use of Genomic Technologies.  The Hastings Center Report 2018 

Reviewer 2 Report

Thank you for your interesting work. I was happy to review it, and have a few suggestions.

- Introduction: in the first paragraph "This expansion has also necessitated consideration..", does not seem to follow logically from the lines before. They illustrate a trend the other way around: new technolgies emerged (MS/MS and now DNA-based), and they have enabled expansion of NBS. https://link.springer.com/article/10.1007/s12687-020-00488-y

- Introduction second paragraph: what do you mean by "candidate disesase" in the last sentence? It is included as a routine disease in many NBS programs now if I'm correct? Perhaps put "candidate" between brackets ().

- Introduction: it might be good to give a rationale for the focus on IEI vs. "traditional" metabolic conditions to make a stronger case for your focus on IEI? For example - if you agree - that you are expecting NGS-based technologies to contribute to IEI to a much greater extent than for metabolic disorders, for which it could be expected that biochemical is and will stay a preferred method?

- " Up-front testing strategy": would first tier testing be more clear - if that is what you mean? Or could you define what you mean by "up-front".

- Table 1: would it be feasible to include what was used as a confirmatory test? Or how disease was confirmed? Electronic patient records, biochemical test etc. FP, TP and other test characteristics are sometimes not as comparable / straightforward as they seem.

- I think it is great that you put a focus on ELSI! Before you dive into the ELSI, could you maybe also add a reflection on the test performance of IEI NBS panel that you mention on page 6? A clinical approach that has been used in NBS: how did the with-symptoms (patient) panel translate to a no-symptom (NBS) population? Or do you consider that included in the comment that WGS panels and methodological option will evolve over time?

> I see some of the points I made above are addressed in 2.2. Maybe it's better to move that section into the introduction or background. I think it illustrates the context more that the actual ELSI?

- The distinction you make seems to line up with the division made by Goldberg et al? Could be good to reference their work. Including ELSI research questions in newborn screening pilot studies - ScienceDirect

- 2.3: the reference to PGx - while I understand it is an opportunity to include in NBS - seems a bit in conflict with your statement "We advocate that only those genes in which mutations are associated with clinically actionable diseases be included in newborn screening panels20". So I would leave PGx for the Discussion.

- 2.3: second paragraph, line 2-3: double "be". And I'm not sure what the " potentially cord blood" adds to the argument? Probably could leave it out or add a reference?

- 2.3 second paragraph: what is meant by " the information", it seems like this also includes late-onset conditions or false positive results? Why would there be poorer parental care? 

- Page 9 second paragraph: could you explain why the fact that a newborn cannot sign a consent form makes it "no basis for requiring outright consent"? The parents can be informed and give consent?

- In 2.4 it seems the argument is missing that NBS programs are well thought out by governments with extensive advice from experts and involvement of many stakeholders. There are many safeguards in place to ensure only relevant testing is done.

- Conclusion: the focus of this manuscript wasn't on the technical aspects of NGS-based NBS for IEI, so saying that it is " an ideal approach" without a reference might be bit strong for the conclusion. Maybe " it is expected to be a suitable approach for IEI with benefits beyond the traditional biochemical test methods" ?

- Conclusion and 2.4: I am not a legal expert on informed consent, but I am unsure if it is true that informed consent isn't needed legally. But that might just be different in different countries as well, maybe this reference says something about it https://pubmed.ncbi.nlm.nih.gov/33040093/ 

Round 2

Reviewer 1 Report

The authors have addressed many of my concerns and suggestions in this revision.  One remaining concern is that the authors have not provided sufficient background on IEI conditions or examples of what they think are good candidates for newborn screening.  It is noted that many of these conditions are monogenic but many monogenic conditions, like CF, have thousands of pathogenic mutations identified, many of which are unique to an individual.  This complicates second tier genetic testing for CF as part of newborn screening, leading to the development of a panel of mutations with the highest frequency.  However, this approach may decrease sensitivity in some populations that are not of northern European ancestry -- an obvious equity issue.  The point is that the manuscript would benefit from the expertise of the authors in the IEI domain.  Perhaps a table would be illustrative that included several IEI conditions the authors think might be suitable for newborn screening.  The table might include the genes responsible, the target variations and their frequency, and perhaps known subtypes of the conditions that are late-onset or subclinical variants.
